Effect of cotton residues incorporation on soil properties, organic nitrogen fractions, and nitrogen-mineralizing enzyme activity under long-term continuous cotton cropping

Ma Fangxia
Wang Yiyun
Yan Peng
Wei Fei
Duan Zhiping
Yang Zhilan
Liu Jianguo l-jianguo@126.com
The Key Laboratory of Oasis Ecology Agriculture of Xinjiang Bingtuan, Shihezi University , Shihezi , Xinjiang , China
Joshi Sanket
Electronic publication date: 2021 Apr 7
Publication date: 2021
Volume: 9
Electronic Location ID: e11053
Received 2020 Jul 9; Accepted 2021 Feb 11
Copyright: ©2021 Ma et al.
Copyright year: 2021
Copyright holder: Ma et al.
License: This is an open access article distributed under the terms of the Creative Commons Attribution License, which permits unrestricted use, distribution, reproduction and adaptation in any medium and for any purpose provided that it is properly attributed. For attribution, the original author(s), title, publication source (PeerJ) and either DOI or URL of the article must be cited.
License URL: https://creativecommons.org/licenses/by/4.0/

Keywords: Cotton residues incorporation, Organic N fractions, N-mineralizing enzymes, Continuous cotton cropping

Funding: National Natural Science Foundation of China 31960396 31560375 This study was supported by the National Natural Science Foundation of China (No. 31960396, 31560375). The funders had no role in study design, data collection and analysis, decision to publish, or preparation of the manuscript.

==============================
The objective of this experiment was to study the effect of cotton residues incorporation on soil properties, soil organic nitrogen (N) fractions, and N-mineralizing enzyme (protease, and urease) activity in the 0–40 cm soil layer in the long-term continuous cotton field. In this experiment, seven treatments, including cotton residues incorporation for 5, 10, 15 and 20 years (marked as 5a, 10a, 15a, and 20a) and continuous cropping for 5, 10 and 20 years (marked as CK5, CK10 and CK20) were conducted. The results showed that the soil organic carbon (C) and N increased gradually with the increase in the duration of continuous cropping with cotton residues incorporation. Compared with CK20, the 20a treatments reduced the content of amino acid N (AAN), ammonium N (AN), amino sugar N (ASN), hydrolysable unidentified N (HUN), and acid insoluble N (AIN) significantly by 48.6, 32.2, 96.9, 48.3, and 38.7%, respectively (p < 0.05). The activity of protease and urease in 20a treatments significantly increased by 53.4 and 53.1% respectively as compared to CK20 (p < 0.05). Soil organic C and N-mineralizing enzyme activity decreased with the increase in cropping duration in the absence of cotton residues incorporation, while the organic N increased slightly. In conclusion, cotton residues returning can increase the storage of soil organic C and N in long-term continuous cropping cotton field, and improve the soil quality and soil fertility of continuous cropping cotton field.

Introduction

Nitrogen (N) is an essential nutrient for crop growth and net primary productivity (Mulvaney, Khan & Ellsworth, 2009). As over 90% soil N is existed in organic forms, soil N availability was primarily determined by soil organic carbon (C) and N (Stevenson, 1982). Crops primarily take up inorganic N and can take in a small part of low molecular weight organic N under extreme conditions (Ashton et al., 2010). Therefore, the mineralization of organic N is a critical parameter regulating ecosystem productivity (Keuper et al., 2017).

The mineralization and depolymerization of organic N in the soil involves a sequence of microbial enzymatic processes (Mengel, 1996). Most of the N input into the soil is in the form of polymers, which must first be decomposed into smaller units by extracellular enzymes (Schimel & Bennett, 2004), which release small organic molecules that can then be directly absorbed or go on degraded, N is absorbed in the form of ammonium (NH4+). Proteases are the most important extracellular depolymerases for the hydrolysis of N-containing molecules, and large proteins and peptides can be hydrolyzed into peptides and amino acids, its activity is closely related to microbial activity (Geisseler & Horwath, 2008). Urease, which released ammonia from linear amides is also an important depolymerase.

As an important source of soil organic matter (SOM), crop residues are rich in C, N, P, K, and trace elements (Malhi, Nyborg & Solberg, 2011a; Malhi, Nyborg & Solberg, 2011b). The incorporation of crop residues into the soil is an important option for improving soil fertility. Bakht, Shafi & Jan (2009) found that crop residue management increased soil N, it helps maintain farmland fertility, and reduce fertilizer utilization. Govaerts, Mezzalama & Unno (2007) also found that residue management improved soil microbial biomass and catabolic diversity.

Xinjiang is the primary cotton cultivation region in China. The perennial, continuous cropping of cotton and full incorporation of cotton residues into the field have become the primary means of organic fertilization in this region. Malhi, Nyborg & Solberg (2011a) and Malhi, Nyborg & Solberg (2011b) found that cotton residues incorporation could increase the content of organic N under conventional and conservational tillage conditions and could also reduce N losses through microbial N sequestration (Shan & Yan, 2013). However, little research has been done to elucidate the response of soil organic N pools and N-mineralizing enzyme activity for long-term continuous cropping and cotton residues incorporation. The objective of this experiment was to study the effects of cotton residues incorporation on soil properties, soil organic N fractions and the relationship between soil organic N fractions and N-mineralizing enzyme activity in a long-term continuous cotton cropping system. We postulated that cotton residues incorporation contributes to the increase in the storage of soil organic C and N in long-term continuous cropping cotton field.

Materials and Methods

Study sites and experimental design

This experiment was conducted at a long-term continuous cropping experimental field at the Shihezi University Agriculture College experimental station (86°03′E, 45°19′N) situated in Shihezi City, Xinjiang Uyghur Autonomous Region, China. The soil texture is gray desert soil, and the basic soil properties are shown in Table 1.

Table 1 Basic soil properties at experimental site.

Duration	Bulk density	pH	Organic matter	Alkali-hydrolyzable nitrogen	Available phosphorus	Available potassium	
(a)	(g cm−3)		(g ⋅kg−1)	(mg ⋅kg−1)	(mg ⋅kg−1)	(mg ⋅kg−1)	
5	1.37	7.31	15.28	70.84	40.8	183	
10	1.38	7.24	16.32	74.36	23.1	110.3	
15	1.35	7.35	17.98	84.06	25.4	65.5	
20	1.33	7.26	18.51	89.08	20.1	72.3	
CK5	1.31	7.29	14.35	113.1	36.47	125.4	
CK10	1.36	7.33	15.67	98.75	46.28	112.8	
CK20	1.34	7.37	15.22	109.4	40.34	108.5	

There were seven treatments in the continuous cropping plots; four of which were cotton residues incorporation treatments, the others were cotton residues removed treatments. There were three replicates per treatment. The experimental design is single factor completely random design. The duration of continuous cropping with cotton residues incorporation treatments included 5, 10, 15, and 20 years, which were marked as 5a, 10a, 15a, and 20a, respectively. The continuous cropping without residues incorporation treatments included 5, 10, and 20 years, which were marked as CK5, CK10, and CK20, respectively. The mode of cotton residues returning to the field for continuous cropping by cutting all the cotton residues into 5–8 cm with a guillotine after harvesting every autumn, namely, simulate the way of returning the cotton residues to the field mechanically, turn it into the plough layer before winter, and then irrigate it in the winter. The pattern of continuous cropping without cotton residues incorporation is to take all the cotton residues out of the field after harvesting, and then apply chemical fertilizer, ploughing, winter irrigation. The cotton was hand harvested. The average contents of C, N, P and K in cotton residues were about 41.3%, 1.69%, 0.43% and 3.14% (by dry weight), respectively. Nitrogen 300 kg hm−2, phosphorus (P2O5) 150 kg hm−2, potassium (K2O) 75 kg hm−2, 30% of N fertilizer and 100% of phosphorus and potassium fertilizer were applied with tillage after cotton harvest. The other 70% N fertilizer was applied with water irrigating. The cultivar of cotton, Xinluzao 46, was sowed in April. On July, spraying growth regulator and manual topping are used to control excessive growth, prevent lodging, increase earliness, and decrease cotton bollworm infestations in cotton field. The way of planting was according to “30+60+30” configuration, using drip irrigation under mulch. The date of sowing was 20th April, and spraying growth regulator and manual topping were on 10th July every year. During the growth period, we would drip 11 times, the total drip was 5,400 m3 hm−2. Other field management followed the local practice.

Soil sampling

Soil samples in the 0–40 cm soil layer were collected in April 2017, five sampling points were combined to form one sample following the “S” shape in each experimental plot. After removing the visible plant roots and stones, the collected soil samples were mixed and passed through a 2 mm sieve. One portion of the samples were stored at 4 °C in order to measure the soil microbial biomass C (Cmic), N (Nmic), and enzyme activity. The other soil samples were air dried and ground through a 0.149 mm sieve so as to determine the content of soil organic C, total C, total N, and organic N fractions.

Soil analyses

Soil organic carbon (SOC) content was determined according to the method of Nelson & Sommers (1982). The content of total organic carbon (Ctot) and total nitrogen (Ntot) in soil was measured by K2Cr2O7 oxidation method and Kjeldahl method. The Soil Cmic content and soil Nmic content were measured by chloroform fumigation extraction method (Joergensen, 1996).

Soil organic N fractions and N-mineralizing enzyme activities

The fractions of organic N in soils were determined by hydrolyzing soil samples with 6 M HCl on an electric hot plate at 120 °C ± 3 °C for 12 h (Bremner, 1965). The content of total hydrolysable N, amino acid N, and hydrolysable ammonium N was measured by the method of steam distillation. The Eqs. (1) and (2) was used to calculate the content of hydrolysable unidentified N and acid insoluble N, respectively. The activities of soil protease, and urease activities were measured by using colorimetrical methods. For the measurement of soil protease activity, 4 g of soil samples, 20 mL 20% casein solution, and 1 mL methylbenzene was added into 50ml glass triangular bottle and incubated for 24 h at 30 °C. Then the solution was mixed with sulfuric acid and sodium sulfate solutions to precipitate protein. After centrifuging, 2ml of the supernatant and ninhydrin solution mixed in the water bath kettle at 100 °C for 10 min and finally determined at 500 nm. For the measurement of urease activity, 5g of soil samples and 1ml methylbenzene were added into 100 mL volumetric flask and incubated for 15 min, then 10 ml 10% urea solution and 20 mL citrate buffer solution (pH 6.7) was added. After the incubation for 24 h, the 38 °C of hot water was added to 100 mL scale line. Then the 3 mL of filtered soil solution was mixed with distillated water, sodium phenoxide, and sodium hypochlorite solution. After 20 min, the mixed soil solution was diluted to 50 mL and determined at 578 nm. (1) Hydrolysable unidentifiedN=total hydrolysableN−amino acidN−ammoniumN−amino sugarN

(2) Acid insolubleN=soil totalN−total hydrolysableN

Statistical analysis

All data were related as means plus one standard deviation. One-way analysis of variance was used to examine the difference between treatments, and the significant difference was considered to reach the p < 0.05 level. Pearson correlation analysis was used for showing the relationship between soil organic N distribution and N-mineralizing enzyme activities. All statistical analysis was carried out by SPSS 19.0.

Results

Soil properties

As shown in Table 2. The contents of Cmic, Nmic, SOC, Ctot and Ntot increased gradually with the increase in continuous cropping duration basically. All of these indexes peaked at 20 years. The Cmic of 20a increased 80.7%, 18.4%, and 36.4%, respectively, in comparison to 5a, 10a, 15a; Nmic increased 92.1%, 20.9%, and 37.2%, respectively; Ctot increased 22.2%, 17.5%, and 6.6%, respectively; Ntot increased 36.1%, 15.6%, and 8.4%, respectively; and SOC increased 44.7%, 13.6%, and 16.5%, respectively; all of which were significantly increased (P < 0.05). However, (C:N)mic and (C:N)tot tended to decrease with the increasing duration of continuous cropping, but no significant difference was observed among treatments. The contents of Cmic, Nmic, Ctot, SOC, and (C:N)tot decreased with the increase in continuous cropping duration, while the contents of Ntot and (C:N)mic increased.

Table 2 Changes of soil properties in 0–40 cm soil depth after long-term continuous cropping and cotton residues incorporation.

Treatment	Cmic	Nmic	Ctot	Ntot	SOC	(C:N)mic	(C:N)tot	
	mg kg−1		g kg−1					
5 a	83.88 ± 5.21d	18.5 ± 0.83d	10.25 ± 0.79c	0.51 ± 0.02d	3.44 ± 0.51c	4.57 ± 0.36ab	20.21 ± 3.21a	
10 a	127.54 ± 8.35b	29.41 ± 2.61b	10.66 ± 3.51c	0.60 ± 0.02c	4.38 ± 0.67b	4.36 ± 0.78ab	17.87 ± 3.71b	
15 a	110.75 ± 8.26c	25.9 ± 4.48c	11.75 ± 3.67b	0.64 ± 0.03b	4.27 ± 1.02b	4.29 ± 1.15ab	18.48 ± 2.57a	
20 a	151.04 ± 10.38a	35.55 ± 6.50a	12.52 ± 2.16a	0.69 ± 0.02a	4.98 ± 0.77a	4.19 ± 0.22b	18.19 ± 5.10ab	
CK5	80.52 ± 3.67d	17.82 ± 3.33d	8.73 ± 1.61d	0.40 ± 0.04g	3.11 ± 0.58cd	4.54 ± 0.73ab	21.7 ± 4.18a	
CK10	77.16 ± 4.88de	16.94 ± 6.20de	8.59 ± 2.30de	0.43 ± 0.02f	2.93 ± 0.60d	4.58 ± 0.80ab	19.96 ± 1.15a	
CK20	67.09 ± 5.12e	14.41 ± 3.52e	7.69 ± 2.23e	0.48 ± 0.04e	2.54 ± 0.39e	4.71 ± 0.54a	16.18 ± 1.06b	
Notes.

Sample size n = 3. Value are means ± standard errors. Upper and lower case indicated significant differences (P < 0.05) between treatments. 5 a: Cotton residues have been incorporated for 10 years in long-term continuous cropping cotton field; 10 a: Cotton residues have been incorporated for 10 years in long-term continuous cropping cotton field; 15 a: Cotton residues have been incorporated for 15 years in long-term continuous cropping cotton field; 20 a: Cotton residues have been incorporated for 20 years in long-term continuous cropping cotton field.

Compared with the treatments without cotton residues incorporation, the content of Cmic, Nmic, SOC, Ctot and Ntot in the treatments with cotton residues incorporation was significantly higher. Compared with CK5, CK10, and CK20, Cmic increased 4.2%, 65.3%, and 125.1%, Nmic increased 3.8%, 73.6%, and 146.7%, SOC increased 10.6%, 49.5%, and 95.9% in 5a, 10a, and 20a, respectively, and with the exception of 5a, these values were all significant (P < 0.05). Additionally, the content of Ctot increased 17.4%, 24.1%, and 62.8%, and that of Ntot increased 26.7%, 39.4%, and 44.4%, respectively (P < 0.05). Cotton residues incorporation thus had no significant effect on (C:N)mic and (C:N)tot.

Soil organic N fractions

The contents of different fractions of soil organic N are shown in Table 3. Under the condition of cotton residues incorporation, amino acid N and amino sugar N, ammonium N, hydrolysable unidentified N, and acid-insoluble N increased gradually with increased continuous cropping basically. All of these indexes reached their maximum value after 20 years of continuous cropping. In 5a, 10a, and 15a, the amino acid N content at 20 years of continuous cropping increased 52.9%, 13.0%, and 23.8%, respectively; ammonium N increased 34.5%, 16.4%, and 8.3%, respectively; amino sugar N increased 42.2%, 11.3%, and 21.9%, respectively; hydrolysable unidentified N and acid-insoluble N increased 32.1%, 15.1%, −3.8%, and 28.9%, 18.4%, and 8.7%, respectively. The difference between treatments were significant (P < 0.05). In the absence of cotton residues incorporation, amino acid N and ammonium N. Hydrolysable unidentified N and acid-insoluble N also increased with the increase in continuous cropping duration, but amino sugar N decreased gradually, the difference was not significant. Amino acid N, ammonium N, amino sugar N, hydrolysable unidentified N, and acid-insoluble N in the treatment with cotton residues incorporation were significantly higher than that of the treatment without cotton residues incorporation. Compared with CK5, CK10, and CK20, the amino acid N of 5a, 10a, and 20a increased 26.87%, 53.33%, and 48.57%; ammonium N increased 8.7%, 7.7%, and 28.6%; amino sugar N increased 11.1%, 53.3%, and 96.9%; hydrolysable unidentified N increased 32.4%, 43.9%, and 48.3%; and acid-insoluble N increased 29.5%, 34.5%, and 38.8%, respectively.

Table 3 Concentrations of N (mg kg-1) in the various fractions in long-term continuous cropping cotton field under straw incorporation.

Treatment	Hydrolysable N	Insoluble N	
	Amino acid N	Ammonium N	Amino sugar N	Unidentified N	Total N		
5 a	99.17 ± 10.11c	84.58 ± 6.90c	26.25 ± 3.86cd	138.25 ± 14.17c	348.25 ± 5.25c	160.32 ± 10.22c	
10 a	134.17 ± 5.05b	97.71 ± 2.53c	33.54 ± 5.05ab	158.67 ± 12.75b	424.08 ± 17.56b	174.49 ± 10.31b	
15 a	122.5 ± 10.11bc	105.00 ± 7.58b	30.63 ± 7.58bc	189.88 ± 6.97a	448.00 ± 16.33b	190.10 ± 12.28ab	
20 a	151.67 ± 10.11a	113.75 ± 4.38a	37.33 ± 3.54a	182.58 ± 15.67a	485.33 ± 12.63a	206.57 ± 12.08a	
CK5	78.17 ± 10.31e	71.46 ± 5.05d	23.63 ± 2.87de	104.42 ± 19.69d	277.67 ± 10.34e	123.70 6 ± 14.99d	
CK10	87.50 ± 5.52cd	80.21 ± 5.05cd	21.88 ± 4.35de	110.25 ± 10.67d	299.83 ± 9.02de	129.69 ± 10.16d	
Notes.

Sample size n = 3. Value are means ± standard errors. Upper and lower case indicated significant differences (P < 0.05) between treatments.

The ratio of amino acid N to total N was 19.5%–22.4%; ammonium N was 16.3%-18.7%; amino sugar N was 4.0%–5.9% (Fig. 1); hydrolysable unidentified N was 25.7%-29.8%; and acid-insoluble N was 29.2%-31.5%, respectively. The organic N fractions were in the following order: acid-insoluble N >hydrolysable unidentified N >amino acid N >ammonium N >amino sugar N. The proportion of amino acid N, acid sugar N, and hydrolysable acid N under cotton residues incorporation increased compared with the treatment lacking cotton residues. Compared with CK5, CK10, and CK20, the proportion of amino acid N increased 0.1%, 10.0%, and 2.9%; acid sugar N increased −12.3%, 10.0% and 36.3%; and the proportion of hydrolysable unidentified N increased 4.5%, 3.3%, and 2.7%, respectively, under continuous cropping for 5, 10, and 20 years. In contrast, the proportion of ammonium N and acid-insoluble N decreased 6.6%, 12.6%, and 8.5% and −2.3%, 3.45%, and 3.9%, respectively, for 5, 10, and 20 years of continuous cropping, compared with CK5, CK10, and CK20. The proportion of amino acid N increased, but the proportion of acid-insoluble N decreased with the increase in continuous cropping duration.

Figure 1 Distribution (%) of soil organic N fractions in long-term continuous cropping cotton field under straw incorporation.

AAN, AN, ASN, HUN, AIN.

Soil N-mineralizing enzyme activities

As shown in Figs. 2 and 3, the protease and urease showed an increasing trend with the increase in cropping duration under the cotton residues incorporation treatment, with values of 18.6 µg g−1 and 550.3 µg g−1, respectively, for 20 years of continuous cropping. Compared with continuous cropping for 5a, 10a, and 15a, the protease activities of continuous cropping for 20 years increased significantly (P < 0.05) 34.2%, 22.5%, and 26.3%, respectively. Urease contents under continuous cropping increased 15.5%, 2.7%, and 1.6%, respectively, and the 5-year treatment showed a significant difference. The activities of the two enzymes all showed a decreasing trend with the increase in continuous cropping duration in the absence of cotton residues incorporation. The enzyme activities under cotton residues incorporation were significantly higher in comparison for the same continuous cropping treatment durations. Compared with CK5, CK10, and CK20, protease activity under continuous cropping for 5a, 10a, and 20a increased 11.8%, 30.4%, and 53.4%, respectively; urease increased 10.3%, 37.6%, and 53.1%, respectively.

Figure 2 Changes of soil protease activities in long-term continuous cropping cotton field under straw incorporation.

Figure 3 Changes of soil urease activities in long-term continuous cropping cotton field under straw incorporation.

Relationship between soil organic N fractions and N-mineralizing enzyme activities

The total organic N and proportion of ammonium N showed significant positive and negative correlations with all N-mineralizing enzymes (protease, urease) (P < 0.05) (Table 4). The proportion of amino acid N, amino sugar N, and hydrolyzed unidentified N were positively correlated with all the N-mineralizing enzymes, while acid-insoluble N was negatively correlated.

Table 4 Correlations between soil organic N distribution (%) and N-mineralizing enzyme activities in the long-term continuous cropping cotton field.

	Protease	Urease	
Amino acid N (%)	0.293	0.157	
Ammonium N (%)	−0.784*	−0.909**	
Amino sugar N (%)	0.513	0.47	
Unidentified N (%)	0.346	0.608	
Insoluble N (%)	−0.546	−0.612	
Total organic N (%)	0.837*	0.854*	
Notes.

*,** statistically significant at P < 0.05 and 0.01, respectively

Discussion

Effect of cotton residues incorporation on soil properties and organic N fractions

The previous study showed that the content of soil total C, N, and SOC in the cotton residues incorporation treatments increased with the duration of continuous cotton cropping increased, and our finding are consistent with Yu et al. (2020). On the contrary, in the treatments that the cotton residues were not incorporated into field, the content of soil total C and SOC decreased with the increasing duration of continuous cotton cropping. This is mainly due to the high organic C content in the cotton residues, which provides exogenous organic matter to the soil and partly enters the soil during microbial decomposition, thus increasing the soil organic C content (Xu, Lou & Sun, 2011). Secondly, because of the high organic C content of the cotton residues, under a high C:N ratio, microorganisms need to absorb more inorganic N from the soil to satisfy their growth requirements, thus improving the ability of microorganisms to utilize ammonium N and nitrate N (Ren et al., 2016). Microorganisms are required for the assimilation of more available N into the soil organic N pool (Tian, Wei & Condron, 2017). Cotton residues incorporation increased the content of C and N in the soil, which indicated the N in the cotton residues incorporation treatment were significantly higher than that lacking cotton residues, which may be due to the increase in soil microbial biomass caused by cotton residues incorporation and the improved soil microbial structure. As the structural index of soil microbial community, the (C:N)mic ratio is used to describe the relative contribution of bacteria and fungi to the soil microbial community, a higher (C:N)mic ratio indicates that the proportion of fungi in the soil microbial community is higher. The lower (C:N)mic ratio indicates that the proportion of bacteria is higher (Tian, Wei & Condron, 2017). In this study, the (C:N)mic ratio decreased gradually with the increased in continuous cropping duration under cotton residues incorporation treatments, while the opposite was observed in the treatments that the cotton residues were not incorporated. The results showed that the number of bacteria in the soil increased gradually with the increase in continuous cropping duration under cotton residues incorporation, which will be beneficial to the development of the cotton field soil in a good direction.

This study also found that cotton residues incorporation increased the content of acid-hydrolyzed ammonium N, and the content increased gradually with the increase in continuous cropping duration. As an important component of soil organic N decomposition, ammonium N also partly originates from soil-fixed NH4+, which constitutes a rapidly released and available soil N pool for plants and microorganisms (Lü, He & Zhao, 2013). The decrease in the proportion of ammonium N to total N under cotton residues incorporation could be attributed to the increased fixation or uptake of mineral N by microorganisms and plants. The acid-insoluble N measured by Bremner’s method is in the form of heterocyclic N or a combination of a heterocyclic compound with an aromatic compound. Heterocyclic compounds and aromatic compounds are stable N compounds. Stable organic N is difficult to be mineralized, leading to its accumulation in the soil (Tian, Wei & Condron, 2017). Secondly, acid-insoluble N is considered as the structural component of humus, and its main source is the senescent substances in aboveground and underground debris (Ren et al., 2016). This can be explained by the increase in humus content in the soil following cotton residues incorporation, leading to the increase in acid-insoluble N content.

Effect of cotton residues incorporation on potential N-mineralizing enzyme activities

Enzymes mainly originate from soil microorganisms and plant secretions in the soil and are the main catalyst of all biochemical processes in the soil. Enzyme activity not only reflects soil quality, but also reflects the direction and intensity of biochemical reactions in the soil (He et al., 2020). It is considered as a sensitive index of soil system change because of the responsiveness, rapid determination, specificity, and comprehensiveness of soil enzymes. Our study showed that cotton residues incorporation could increase the soil urease and protease activities, which was due to the increase in soil urease and protease activities, which was due to the increase in soil microbial C under cotton residues incorporation treatments. On the contrary, the soil microbial C decreased gradually with the increasing duration of continuous cropping when the cotton residues were not incorporated, leading to the decreased soil enzyme activity.

Potential mechanisms involved in soil organic N turnover under the condition of cotton residues incorporation

The transformation and absorption of soil N is conducted by enzyme systems, and their synthesis and expression require C, N, and energy, suggesting that the form of available N, the source of C, and the C availability in relation to N are important factors (Yang, Zhang & Geisseler, 2016). It is generally believed that all organic N is mineralized to NH4+ before being absorbed by soil microorganisms. This pathway is generally known as the mineralization-immobilization-turnover (MIT) route. When the concentration of NH4+ was relatively high (Fig. S1), the gene transcription of the enzyme systems needed to replace the N source were inhibited, and cell N-mineralizing enzymes were involved in the potential transformation of soil organic N under the condition of cotton residues incorporation, enzymatic mineralization is superior to the direct route (Tian, Wei & Condron, 2017). Under C-limited conditions, the substrates of N with low molecular weights, such as amino acids and amino sugars, can be directly absorbed by microorganisms as a C source (Nannipieri & Eldor, 2009). There were significant positive correlations between soil total organic N and N-mineralizing enzyme activities, indicating that the activities of N-mineralizing enzymes in the soil played an active role in the potential transformation of soil organic N. However, there was no significant positive correlation between the ratios of amino acid N, amino sugar N, and acid-hydrolyzed unknown N and the activity of N-mineralizing enzymes, which was due to the fact that the transformation of soil organic N was restricted by the lack of C source. But the ratio of ammonia N showed a significant negative correlation with the activity of N-mineralizing enzymes. This is because the increased soil N-mineralizing enzymes after cotton residues incorporation promoted the mineralization of soil organic N.

In the cotton residues incorporation treatments, the decrease of the proportion was attributed to the increase of fixation or absorption of mineral N by microorganisms and plants. Therefore, in the soil of long-term continuous cropping cotton field, soil microorganisms need to absorb organic N molecules to meet the demand for C due to cotton residues incorporation.

Conclusions

Cotton residues incorporation was beneficial to improve soil quality and soil fertility in the long-term continuous cotton cropping field. The benefits can be increased over time. Under the condition of cotton residues incorporation, when the concentration of NH4+ was high, enzymatic mineralization constituted the main pathway of potential organic N turnover. However, when cotton residues were not incorporated into the soil, the utilization of soil organic N was the most direct route due to the lower soil organic C availability.

Supplemental Information

Data S1 Data

Data from Figs. 1–2 and Tables 1–3.

Click here for additional data file.

Figure S1 Changes of ammonia nitrogen content in 0–40 cm soil layer in long-term continuous cropping cotton field

Click here for additional data file.

Additional Information and Declarations

Competing Interests

Author Contributions

Data Availability

The authors declare there are no competing interests.

Fangxia Ma conceived and designed the experiments, performed the experiments, analyzed the data, prepared figures and/or tables, authored or reviewed drafts of the paper, and approved the final draft.

Yiyun Wang and Peng Yan performed the experiments, analyzed the data, prepared figures and/or tables, and approved the final draft.

Fei Wei and Zhiping Duan performed the experiments, prepared figures and/or tables, and approved the final draft.

Zhilan Yang performed the experiments, prepared figures and/or tables, and approved the final draft.

Jianguo Liu conceived and designed the experiments, prepared figures and/or tables, authored or reviewed drafts of the paper, and approved the final draft.

The following information was supplied regarding data availability:

Raw data are available in a Supplemental File.

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
