# Peer review of "Effect of cotton residues incorporation on soil properties, organic nitrogen fractions, and nitrogen-mineralizing enzyme activity under long-term continuous cotton cropping"

_PeerJ, doi:10.7717/peerj.11053_

## Round 0.1 · original submission · Major Revisions

Please go through the reviewer's constructive comments, including about some language editing issues, discussions and figures. Kindly revise your manuscript considering those comments and resubmit your manuscript.

Reviewer 1 ·

Basic reporting

1. There are some grammar errors through the manuscript, authors should improve the wriiten English. If available, please ask a native speaker to help.
2. Not enough background and knowledge gap, authors should improve the introduction section.
3. The structure, figures and tables are fine.
4. Last but not least,nitrate reductase, and nitrite reductase are not N-mineralizing enzymes. This makes this manuscript not suitable to be published.

Experimental design

The experiment was designed reasonable. However, the authors classified nitrate reductase, and nitrite reductase into wrong group.

Validity of the findings

The results are meaningful to understand how carbon and nitrogen change after straw return. But from the title, the results of carbon took too much pages. "Soil N-mineralizing enzyme activities " included two unrelated enzymes. The wrong classification of enzymes directly resulted wrong conclusions.

Additional comments

1. Please note that nitrate reductase, and nitrite reductase are not N-mineralizing enzymes. They are denitrification enzymes.
2. Try to find a native speaker to improve the written English.
3. The disscussion section should be clearer, it is hard to read now.
4. Try to include carbon in the title or delete some results on carbon.
5. Please re-draw figure 2. Do not merge the sub-figures.

·

Basic reporting

The manuscript is well organized and unambiguous. Language used in the manuscript is up to professional standards. Introduction is oriented in towards specific goals of the study. Results are structured properly with matching figures and tables. Manuscript is written with proper references.

Experimental design

Experimental methods are thoroughly described and well within standards of the field. Methods used in the investigation are well known in the field to address the questions of the agriculture related study. Overall the investigation is supported by well documented methods.

Validity of the findings

Conclusions drawn in the study are supported by results described in the manuscript. Conclusions are made from statistical significant results.

Additional comments

The manuscript entitled “Effect of cotton residues incorporation on soil organic nitrogen fractions and nitrogen-mineralizing enzyme activity under long-term continuous cotton cropping” shows interesting results and presents important agricultural study of soil organic nitrogen, organic carbon and nitrogen-mineralizing enzyme activities.

Authors are requested to explain following results.

Line 245-246: The numbers described for protease, nitrate reductase and nitrite reductase do not match those described in Figure 2.

Figure 2: Significant increase in 15a compared to 10a has not been observed for protease, urease, nitrate reductase and nitrite reductase activities. While significant increase has been observed for 10a and 20a compared to 5a for all enzyme activities except nitrate reductase. Is there a specific reason for this discrepancy for 15a results?

Typological corrections:
Figure 2: Two panels are labelled as Protease activities. On of them has to be for nitrite reductase.

---

## Round 0.2 · Minor Revisions

The revised version reads much clear and better than the previous one. However, as highlighted by the reviewer it still needs some minor corrections. Kindly do the needful and submit the revision.

Reviewer 1 ·

Basic reporting

The written English is clear and professional English used throughout the article. However, more recent literatures (recent 5 years) should be cited to replace old references.

Experimental design

The experimental design was reasonable and could support the research goal.

Validity of the findings

There are some new findings in the result. Conclusions are clear and related to objectives.

Additional comments

1. Please verify why 0-40 cm soil layer was selected in this study。
2. Please include important numbers in the abstract.
3. The methods to analyze N-mineralizing enzyme activities should be stated more, only one sentence is not enough to gain efficient knowledge.
4. Authors should note that the year when the experiment was started should be listed and the basic soil properties were lacked, please supply these data.
5. Try to discuss with recent literatures, i.e., published from 2016.

·

Basic reporting

Manuscript language has been improved with review.

Experimental design

Necessary changes in experimental results have been made according to suggestions from reviewers. Reviewed manuscript fits desired objectives of the investigation.

Validity of the findings

No comments

Additional comments

No comments

---

## Round 0.3 · accepted · Accept

Authors corrected the manuscript considering all suggestions and comments, and the revised version reads much better and clear now.